# Association between timed up-and-go test and subsequent pneumonia: A cohort study

Hyo Jin Lee[1], Sohee Oh[2], Hyun Woo Lee[1], Jung-Kyu Lee[1], Eun Young Heo[1], Deog Kyeom Kim[1,3], Tae Yun Park[1] *

1 Division of Respiratory and Critical Care, Department of Internal Medicine, Seoul Metropolitan Government-Seoul National University Boramae Medical Center, Seoul, South Korea, 2 Medical Research Collaborating Center, Seoul Metropolitan Government-Seoul National University Boramae Medical Center, Seoul, South Korea, 3 Department of Internal Medicine, Seoul National University College of Medicine, Seoul, Republic of Korea

* zinhinim@gmail.com

## Abstract

### Background

Sarcopenia is a risk factor for pneumonia in the elderly, and the timed up-and-go test (TUG) can be used as a screening tool for sarcopenia in this population. This study aimed to evaluate the association between TUG test results and future pneumonia or ventilator care.

### Materials and methods

From the National Health Insurance Service-Senior Cohort database, we identified 19,804 people without neurological diseases who underwent the TUG test in the National Screening Program for Transitional Ages at the age of 66 years during 2007–2008. Gait abnormality was defined as taking 10 s or longer to perform the TUG test. Pneumonia occurrence was defined using the International Classification of Diseases 10th Revision (ICD-10) code for pneumonia (J12–J18, J69), and ventilator care was defined by procedure codes (M5830, M5850, M5867, M5858, M5860, M5859) according to the Healthcare Common Procedure Coding system codes from 2007 to 2015.

### Results

The mean follow-up period was 7.4 years (standard error, SE 0.02). The incidence rates of pneumonia in the normal and slow TUG groups were 38 and 39.5/1000 person-years, respectively. The slow TUG group did not show a higher risk of pneumonia (adjusted hazard ratio [aHR], 1.042; 95% confidence interval [95% CI], 0.988–1.107]). Regarding ventilator care, the incidence was 4.7 and 5.2 cases per 1,000 person-years in the normal and slow TUG groups, respectively. Slow TUG groups also did not show an increased risk of ventilator occurrence (aHR, 1.136, [95% CI = 0.947–1.363]).

### Conclusion

The TUG test result was not associated with future pneumonia or ventilator care and may not be useful for predicting pneumonia in community-dwelling elderly individuals. Further

**Data Availability Statement:** Data cannot be shared publicly because of the health screening cohort database of the Korean National Health Insurance Service (NHIS). Data are available from the National Health Insurance Service (NHIS)

Institutional Data Access / Ethics Committee (contact via NHIS) for researchers who meet the criteria for access to confidential data. Contact details: URL: https://nhiss.nhis.or.kr/; E-mail: nhiss@nhis.or.kr; Phone: +82-1577-1000; Address: 32 Gungang-ro, Wonju-si, Gangwon-do 26464.

**Funding:** The authors received no specific funding for this work.

**Competing interests:** The authors have declared that no competing interests exist.

**Abbreviations:** aHR, adjusted hazard ratio; BMI, body mass index; CCI, Charlson comorbidity index; CI, confidence interval; GDS, Geriatric Depression Scale; ICD-10, International Classification of Diseases 10th Revision; KDSQ-C, Korean Dementia Screening Questionnaire; KNHI, Korean National Health Insurance; NHIS-Senior, National Health Insurance Service-Senior Cohort; NHSP, National Health Screening Program; NSPTA, National Screening Program for Transitional Ages; PA, physical activity; SD, standard deviation; TUG, Timed Up-and-Go test; UBT, unipedal balance test.

studies are needed to identify additional functional tools for sarcopenia associated with future pneumonia occurrences.

## Introduction

Pneumonia is one of the leading causes of death worldwide, particularly in elderly patients [1–3]. This is due to the growing elderly population [4], who are more vulnerable to pneumonia because of their impaired gag and cough reflex, decreased mucociliary function, and decreased swallowing response [3, 5].

With aging, the loss of muscle mass and function gradually progress to sarcopenia [3, 6, 7]. Sarcopenia may also disrupt the aforementioned protective mechanisms in the elderly population, eventually leading to pneumonia.

Previous studies have demonstrated an association between sarcopenia and pneumonia. These studies found that weakened handgrip strength [8] and decreased calf circumference [9, 10] were risk factors for pneumonia and that sarcopenia is associated with the mortality rate of pneumonia. These findings suggest that the diagnostic tests available for sarcopenia may be useful for evaluating the risk of pneumonia in elderly individuals.

Among the various tests and equipment for detecting sarcopenia [11–13], the timed up-and-go (TUG) test is a simple, inexpensive, and feasible assessment tool for sarcopenia [14, 15]. It uses the time that a person takes to rise from a chair, walk 3 m, turn around 180˚, walk back to the chair, and sit down while turning 180˚ [15]. Since this test evaluates basic functional sarcopenia in frail elderly persons, it may be useful for detecting the risk of pneumonia occurrence [15].

Numerous studies to support that the TUG test can be feasible assessment tool for sarcopenia. In a study evaluating the performance of the TUG test as a screening tool for sarcopenia among 332 older adults in a southern Brazilian city, the test effectively screened for sarcopenia with a high sensitivity (88.9%) and negative predictive value (93.2%), particularly those with good physical and cognitive abilities [16]. Another study also demonstrated that TUG test predicted sarcopenia with a sensitivity of 67% and a specificity of 88.7% in elderly patients [15]. Furthermore, a cross-sectional study showed that muscle mass and physical performance, which are assessment tool for sarcopenia, were associated with gait-speed and TUG tests [17]. In this study, elderly women with reduced muscle mass exhibited poor physical performance, as indicated by slow TUG test taking over 10.85 s. In addition, Korean Working Group on Sarcopenia Guideline suggests that TUG test may simplify the diagnostic steps for sarcopenia in clinical settings with a high prevalence of sarcopenia [18].

Given that the TUG test is recognized as a valuable tool for diagnosing sarcopenia, and considering the established link between sarcopenia and an increase risk of pneumonia, we hypothesized that the TUG test could serve as a useful predictive tool for pneumonia occurrence in community-based elderly populations. However, to the best of our knowledge, there is no research on the potential role of the TUG test in predicting the occurrence of pneumonia in elderly people.

Thus, this study aimed to evaluate the association between the baseline TUG test and future pneumonia occurrence in the elderly population using a nationwide study of health insurance claims.

## Materials and methods

### Study setting

The Korean National Health Insurance (KNHI) service is a compulsory health insurance system that covers the universal health of Koreans. The KNHI service provides a biennial

National Health Screening Program (NHSP) to all members over 40 years of age. The NHSP includes a questionnaire (sociodemographic data, past medical history, health behavior), an anthropometric examination (body mass index, blood pressure), and laboratory tests (blood sugar, cholesterol, etc.). Since 2007, the KNHI has also provided a special health-screening program called the National Screening Program for Transitional Ages (NSPTA) for all people aged 66 years. In addition to the routine items of the NHSP, the NSPTA also includes a questionnaire regarding mental/cognitive function and depression screening, as well as physical function tests (TUG test, unipedal balance test) that cover common problems of the elderly, such as frailty.

## Study population

The KNHI database has been extensively used in various epidemiological studies [19], and its details and validities have been described previously. This study used data from the National Health Insurance Service-Senior Cohort (NHIS-Senior) database, which comprises approximately 5,500, 000 individuals aged 60 and older who were still eligible for national health insurance and medical benefits at the end of December 2002. Among the 5,500,000 people in the NHIS-Senior database, approximately 10%, 558, 147 people, were selected using simple random sampling. The cohort was followed-up retrospectively in 2015 for all subjects. The total number of patients in the cohort was 558,147 at the beginning (2002) and 352,869 at the end of the follow-up period (2015). The NHIS-Senior cohort contains demographic factors, such as age, sex, death date, and results of the NHSP, as well as information on the utilization of medical facilities, including the International Classification of Diseases 10th Revision (ICD-10) codes, prescribed medicines from outpatient clinics, and hospitalizations. As NHI is mandatory social insurance in Korea, attrition from this cohort occurs only by emigration, which is not common in those aged over 60 years.

For this study, individuals aged 66 years who underwent the TUG test in the NSPTA program were selected. The initial study population comprised 40,779 participants. The first year (2006) was designated as the washout period. Participants who were registered with severe disability status (n = 38) or diagnosed with disabling diseases, such as stroke (n = 18,836), Parkinson's disease (n = 108), dementia (n = 25), and those with missing records (n = 66) or inaccurate TUG test results (n = 1,528), were excluded. We also excluded participants with pneumonia during the 1-year washout period (n = 580). Ultimately, 19,804 participants were included in the analysis (Fig 1). This study was approved by the Institutional Review Board of Seoul National University Boramae Hospital (IRB No. 07-2020-193). Consent from individual participants was waived because publicly available anonymous data were used.

## Independent variable

The TUG test was performed on the day of the examination of the NSPTA program at community hospitals. Participants were timed while they rose from a chair, walked at a comfortable pace to a line on the floor 3 m away, turned, walked back to the chair, and sat down again [20]. They wore regular footwear and used customary walking aids.

The cut off value of the TUG test varies across studies. but the cut off value of 9–10 s is used in various population groups [20, 21], and the cut off value of 10 s is also used in Asian populations such as Japan and Singapore [22, 23]. In fact, most studies investigating the association of the TUG test with clinical outcomes such as dementia, fractures, and functional disability in Korean populations have used a 10 s cut off value [24–27]. Based on these existing studies and expert opinions, the Korean NSPTA defines a TUG test of less than 10 s as normal. Thus, in

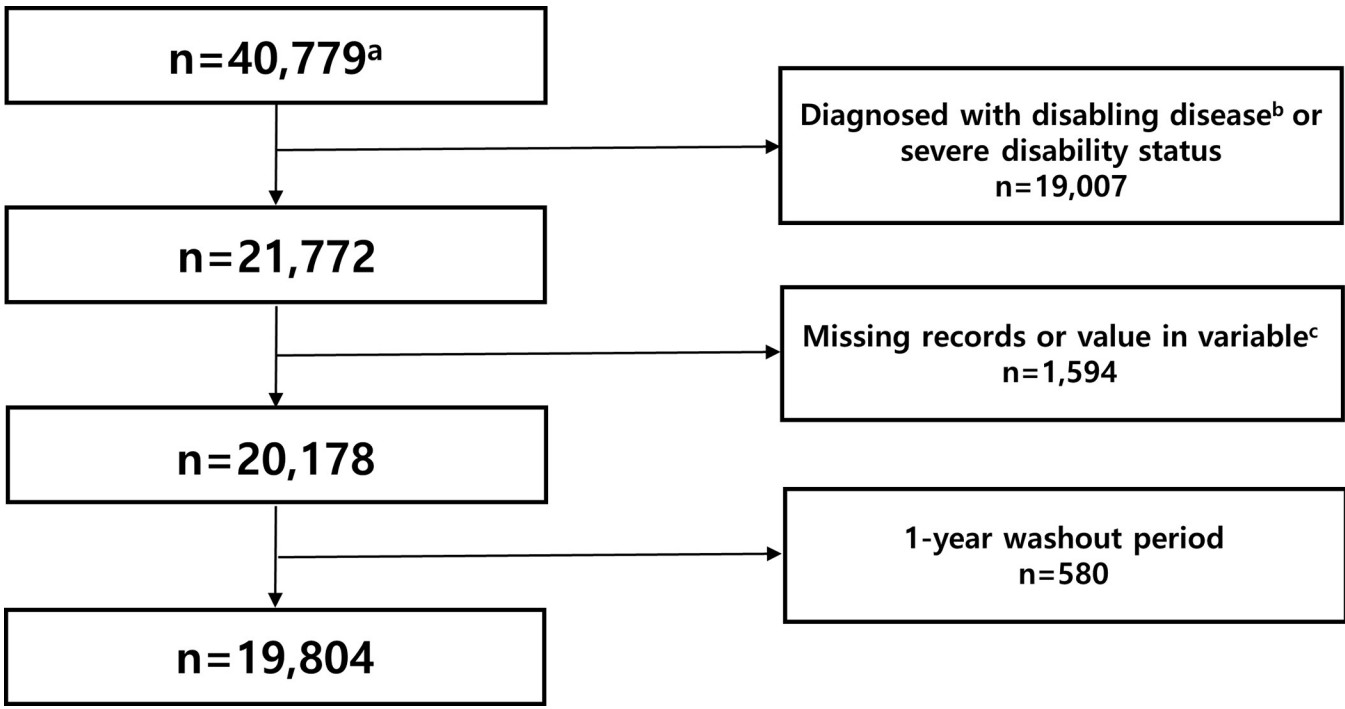

**Fig 1. Flow chart of study population selection process.** [a]Initial population: People 66 years old who underwent TUG test who participated in the National Screening Program for Transitional Ages (NSPTA) during 2002–2015 were extracted from the National Health Insurance Service-Senior Cohort (NHIS-Senior) database. [b]Disabling disease: stroke, Parkinson's disease, dementia. [c]Missing value in variables: Timed up-and-go test. Abbreviations: TUG, timed up-and-go.

our study, participants were divided into two groups based on their TUG test ($<$10 s as normal and $\geq$10 s as slow gait).

## Outcome variable

The primary outcome was the incidence of pneumonia during the follow-up period. Pneumonia was defined according to the ICD-10 codes for pneumonia (J12–J18) and aspiration pneumonia (J69) [28, 29]. Ventilator care was defined as the presence of one of the following procedure codes for ventilator care: M5830, M5850, M5867, M5858, M5860, and M5859, according to the Healthcare Common Procedure Coding System codes provided by the Health Insurance Review and Assessment Service [30, 31].

## Covariates

Sex, smoking, body mass index (BMI, kg/m$^2$), comorbid conditions, depressive symptoms, physical function tests, and baseline cognitive function were considered related to the future development of pneumonia and were included as covariates in the analysis. Information regarding smoking history was obtained from a questionnaire administered on the screening day. The questionnaire includes inquiries regarding the current smoking status of participants: (1) Never smoker, (2) Former smoker who has quit, and (3) Current smoker. Additionally, the questionnaire includes inquiries regarding smoking history, both past and present, with regards to: (1) Smoking duration (1. $<$5 years, 2. 5–9 years, 3. 10–19 years, 4. 20–29 years, 5. 30 years or more), and (2) Daily cigarette consumption (1. $<$0.5 pack, 2. 0.5–1 pack, 3. 1–2 packs, 4. 2 packs or more). Based on the smoking information provided, participants were classified

as never smokers, former-smokers, or current smokers. Data on relevant comorbid conditions were also collected from the questionnaire and claim data under ICD-10 codes before the screening date, which were represented as Charlson Comorbidity Index (CCI) scores. An abnormal unipedal balance test (UBT) was defined as the inability to maintain a posture for 5 s or less with closed eyes or 9 s or less with open eyes [32, 33]. The activities of daily living (ADL) test assessed the individual's ability to independently perform six activities such as bathing, dressing, eating, using the bathroom, preparing meals, and going out to various locations unassisted. ADL impairment was defined as the inability to perform three or more of these items. The history of the falls was based on self-reported information. The Prescreening Korean Dementia Screening Questionnaire (KDSQ-C) was used to evaluate cognitive function, and a KDSQ-C score of ≥6 was considered an abnormal finding requiring further cognitive function assessment. Depressive symptoms were evaluated using a three-item questionnaire extracted from the Geriatric Depression Scale (GDS), including items 2, 17, and 22, which relate to the loss of interest, feelings of uselessness, and feelings of hopelessness, respectively.

## Statistical analysis

Continuous variables are presented as mean ± standard deviation (SD), and categorical variables are shown as numbers and proportions. To compare the clinical and demographic characteristics of the TUG test groups (< 10 s vs. ≥10 s), we performed a two-tailed Student's t-test for continuous variables and a chi-squared test for categorical variables.

The patients were followed-up from the day of screening of the NSPTA program to the occurrence of pneumonia, ventilator care, or the last follow-up day (December 31, 2015), whichever came first. Kaplan-Meier curves and log-rank tests were used to describe the difference between the TUG test groups and event occurrence, including pneumonia and ventilator care. Multivariate analysis was performed using the Cox proportional hazards regression model with possible risk factors for pneumonia. Potential factors were selected as variables with a statistically significant association on univariable analysis, which included sex, BMI, smoking, CCI score or comorbidities, physical function tests (unipedal test and TUG test), regular physical activities, depressive symptoms, and baseline cognitive function. Associations between the TUG test groups (results) and outcome variables were assessed using hazard ratios and 95% confidence intervals. Statistical significance was set at P < 0.05. All statistical analyses were performed using SAS Enterprise Guide version 7.1 (SAS Institute Inc., Cary, NC, USA) and R version 3.3.3 (The R Foundation for Statistical Computing, Vienna, Austria).

## Ethics approval and consent to participate

This study was conducted in accordance with the ethical guidelines of the Declaration of Helsinki in 1975. The Institutional Review Board Committee of Seoul National University Seoul Metropolitan Government (SNU-SMG) Boramae Medical Center approved the study protocol and waived the requirement for informed consent from study subjects for access to electronic medical records (IRB No. 07-2020-193).

## Results

### Baseline characteristics

Among the 19,804 participants who were initially included in the analysis, 60.3% had a normal TUG test (<10 s), and 39.7% had slow TUG (≥10 s) (Table 1). The mean TUG time was 9.5 s (standard deviation [SD] 3.98); the mean TUG was 7.6 s (SD 3.98) in the normal TUG group

**Table 1. Baseline characteristics of the study population.**

| | Total | TUG test | | |
| --- | --- | --- | --- | --- |
| | | <10 s | ≥10 s | P-value |
| Number | 19,804 | 11,949 (60.34%) | 7,855 (39.66%) | |
| TUG time (sec) | 9.52 ± 3.98 | 7.64 ± 1.03 | 12.37 ± 4.99 | <0.0001 |
| Sex, male | 9,076 (45.83%) | 5,778 (48.36%) | 3,298 (41.99%) | <0.0001 |
| Age (years) | 66.06 ± 0.23 | 66.05 ± 0.22 | 66.07 ± 0.25 | <0.0001 |
| BMI (Kg/m$^2$) * | 24.25 ± 3.02 | 24.19 ± 2.93 | 24.34 ± 3.15 | <0.0001 |
| Smoking | | | | <0.0001 |
| Never smoker | 13,036 (73.42%) | 7,675 (71.73%) | 5,361 (75.99%) | |
| Ever-smoker** | 4,719 (26.57%) | 3,025 (28.27%) | 1,694 (24.01%) | |
| Drinking | | | | <0.0001 |
| Non-drinker | 2,341 (30.38%) | 1,911 (36.32%) | 430 (17.60%) | |
| Moderate | 4,327 (56.16%) | 2,720 (51.69%) | 1,607 (65.78%) | |
| Heavy | 1,037 (13.46%) | 631 (11.99%) | 406 (16.62%) | |
| Income (lowest quantile) | 2,560 (12.93%) | 1,553 (13.00%) | 1,007 (12.82%) | 0.7164 |
| Regular PA, yes | 1,4960 (86.46%) | 8,510 (81.16%) | 6,450 (94.63%) | <0.0001 |
| Fall History [a], yes | 2,320 (11.98%) | 1,313 (11.27%) | 1,007 (13.06%) | 0.0002 |
| UBT, abnormal | 3,285 (17.91%) | 1,752 (15.75%) | 1,533 (21.24%) | <0.0001 |
| KDSQ-C, abnormal | 704 (3.65%) | 385 (3.31%) | 319 (4.15%) | 0.0024 |
| Depressive mood | 6,937 (35.79%) | 4,026 (34.50%) | 2,911 (37.75%) | <0.0001 |
| ADL, impaired | 18,868 (97.60%) | 11,332 (97.44%) | 7,536 (97.84%) | 0.0702 |
| CCI score | 0.46 ± 0.71 | 0.45 ± 0.71 | 0.48 ± 0.72 | 0.0133 |
| Comorbidities | | | | |
| Hypertension | 7,970 (40.24%) | 4,770 (39.92%) | 3,200 (40.74%) | 0.2504 |
| DM | 3,359 (16.96%) | 1,919 (16.06%) | 1,440 (18.33%) | <0.0001 |
| COPD | 311 (1.57%) | 177 (1.48%) | 134 (1.71%) | 0.2136 |
| CKD | 58 (0.29%) | 39 (0.33%) | 19 (0.24%) | 0.2817 |
| Heart disease | 1,356 (6.85%) | 802 (6.71%) | 554 (7.05%) | 0.3526 |
| Cancer | 339 (1.71%) | 211 (1.77%) | 128 (1.63%) | 0.4694 |
| CLD | 793 (4.00%) | 471 (3.94%) | 322 (4.10%) | 0.5801 |

**Note**: Data are presented as n (%) or median (SE).

[a] Fall history in the preceding 6 months.

* Continuous variable

** Former or current smokers

**Abbreviations**: ADL, activities of daily living; BMI, body mass index; CCI, Charlson Comorbidity Index; CKD, chronic kidney disease; CLD, chronic liver disease; COPD, chronic obstructive pulmonary disease; DM, diabetes mellitus; KDSQ-C, Korean Dementia Screening Questionnaire; PA, physical activity; TUG, timed up-and-go; UBT, unipedal balance test.

and 12.4 s (SD 4.99) in the slow TUG group. There were more underweight (BMI <18.5 kg/m$^2$, 2.3%) or obese (BMI ≥25 kg/m$^2$, 39.1%) individuals in the slow TUG group than in the normal TUG group. The percentage of women (51.6% vs. 58.0%), never smokers (71.7% vs. 76.00%), and moderate or heavy alcoholics (63.7% vs. 82.4%) was higher in the slow TUG group than in the normal TUG group.

In terms of physical activity (PA) and mental performance, individuals with longer TUG times showed poor geriatric physical performance (a history of falls and unipedal balance test abnormality) and an abnormal mental health state (poor cognitive function or depressive mood), whereas individuals in the normal TUG group had fewer regular PA than those who

**Table 2. Association between risk factors and pneumonia occurrence (n = 19,804).**

| Variables | Univariable analysis | | Multivariable analysis | |
|---|---|---|---|---|
| | HR (95% CI) | P-value | aHR (95% CI) | P-value |
| Sex, Male | 1.14 (1.08–1.21) | < 0.0001 | 1.06 (0.98–1.15) | 0.1622 |
| BMI, kg/m²* | 0.98 (0.97–0.99) | 0.0003 | 0.99 (0.98–0.99) | 0.0133 |
| UBT-abnormal | 1.16 (1.08–1.25) | < 0.0001 | 1.14 (1.05–1.24) | 0.0014 |
| Ever-smoker** | 1.19 (1.11–1.27) | < 0.0001 | 1.12 (1.03–1.23) | 0.0115 |
| Regular PA | 1.10 (1.00–1.21) | 0.0358 | 1.06 (0.96–1.17) | 0.2438 |
| Cognitive function-abnormal | 1.21 (1.05–1.34) | 0.0086 | 1.05 (0.89–1.24) | 0.5447 |
| Depressive mood | 1.11 (1.05–1.18) | 0.0005 | 1.10 (1.03–1.18) | 0.0047 |
| CCI score | 1.35 (1.31–1.40) | < 0.0001 | 1.35 (1.30–1.40) | <0.0001 |

**Note:** Data were analyzed using univariable and multivariable Cox regression models and are presented as adjusted hazard ratios (95% confidence intervals).

\* Continuous variable

\*\* Former or current smokers

**Abbreviations**: BMI, body mass index; CCI, Charlson comorbidity index; PA, physical activity; UBT, unipedal balance test.

[a] Multivariable model included sex, smoking status, body mass index, unipedal balance test, regular physical activity, baseline cognitive function, depressive symptoms, and Charlson comorbidity index.

had longer TUG times. Comorbidities showed a similar distribution between both groups, except for diabetes mellitus (16.1% vs. 18.3%).

## Association between risk factors and pneumonia or ventilator care events

In univariate analysis, there was a significant association between pneumonia occurrence and the following factors: male sex, low BMI, abnormal unipedal balance test, former or current smoking, increased regular PA, abnormal mental and cognitive dysfunction, and high CCI score. The results of the multivariable analysis are shown in Table 2. In the multivariable analysis, low BMI, abnormal unipedal balance test, former or current smoking, depressive mood, and high CCI score showed higher risks of pneumonia occurrence (Table 2).

Regarding ventilator events, the same risk factors for pneumonia occurrence were observed in univariable analysis, except for regular PA and abnormal cognitive dysfunction. Male sex, abnormal unipedal balance test results, and high CCI scores were associated with ventilator use in multivariable analysis (Table 3).

## Association between TUG test and risk of pneumonia or ventilator care events

The mean follow-up duration was 7.4 (SE 0.02) years in the normal TUG group and 7.2 (SE 0.03) years in the slow TUG group. The incidence rates of pneumonia in the normal and slow TUG groups were 38 and 39.7/1000 person-years, respectively. In univariate analysis, the slow TUG group showed a trend of higher risk of pneumonia occurrence compared with the normal TUG group; however, the difference was not statistically significant (hazard ratio [HR], 1.046 [95% CI = 0.988–1.107]). Similarly, in the multivariable analysis, there was no difference in pneumonia occurrence between the slow and normal TUG groups (adjusted HR [aHR], 1.042; [95% CI, 0.988–1.107]) (Table 4). In the analysis of ventilator care events, the incidence rates of ventilator care events were 4.7 and 5.2 cases per 1,000 person-years in the normal and slow TUG groups, respectively. Slow TUG test was not associated with a higher risk of ventilator care events in the univariate (HR, 1.095 [95% CI = 0.942–1.274]) and multivariable analyses (aHR, 1.136; [95% CI, 0.947–1.363]) (Table 4).

**Table 3. Association between risk factors and ventilator events (n = 19,804).**

| Variables | Univariable analysis | | Multivariable analysis | |
|---|---|---|---|---|
| | HR (95% CI) | P-value | aHR (95% CI) | P-value |
| Sex, Male | 2.09 (1.79–2.44) | <0.0001 | 2.00 (1.61–2.50) | <0.0001 |
| BMI* | 0.98 (0.95–0.99) | 0.0442 | 0.99 (0.96–1.02) | 0.4714 |
| UBT-abnormal | 1.38 (1.14–1.67) | 0.0008 | 1.44 (1.16–1.78) | 0.0008 |
| Ever-Smoker** | 1.73 (1.46–2.04) | <0.0001 | 1.16(0.93–1.44) | 0.1971 |
| Regular PA | 0.97 (0.77–1.22) | 0.7708 | 0.83 (0.64–1.07) | 0.1483 |
| Cognitive function-abnormal | 1.06 (0.72–1.57) | 0.7632 | 1.05 (0.66–1.64) | 0.8505 |
| Depressive mood | 1.25 (1.07–1.45) | 0.0050 | 1.19 (0.99–1.43) | 0.0666 |
| CCI score | 1.42 (1.31–1.54) | <0.0001 | 1.42 (1.30–1.56) | <0.0001 |

**Note:** Data were analyzed using univariable and multivariable Cox regression models and are presented as adjusted hazard ratios (95% confidence intervals).

**Abbreviations**: BMI, body mass index; CCI, Charlson comorbidity index; PA, physical activity; UBT, unipedal balance test.

[a] Multivariable model included sex, smoking status, body mass index, unipedal balance test, regular physical activity, baseline cognitive function, depressive symptoms, and Charlson comorbidity index.

* BMI is shown as a continuous variable.

** Former or current smokers

The occurrence of pneumonia and ventilator care during the follow-up period is shown in Fig 2. The cumulative incidence trend graph for pneumonia and ventilator care from 2007 to 2015 is shown in Fig 3.

## Discussion

This study suggests that slow TUG time performance (≥10 s) is not associated with the risk of subsequent pneumonia or ventilator care events. However, risk factors, including abnormal unipedal balance test, male sex, low BMI, depressive mood, former or current smoking status, and high CCI score, were associated with a higher risk of subsequent pneumonia or ventilator care events.

The TUG test is a well-known clinical assessment tool designed to evaluates balance and functional performance in older adults [34]. Because muscle strength and gait-speed are essential factors for functional disability and diagnosing sarcopenia [11–13], the TUG test has been widely used in many studies on the relationship between functional disability and healthcare outcomes in elderly population [25, 27, 35–37]. In a retrospective study involving 974 older

**Table 4. Occurrence of pneumonia according to baseline timed up-and-go test results (n = 19,804).**

| | Person-years | Number of Occurrence | Rate (1/1,000) | HR (95% CI) | aHR (95% CI)[a] |
|---|---|---|---|---|---|
| Pneumonia | 126,066.5 | 4874 | 38.7 | | |
| <10 s | 76,315.1 | 2898 | 38 | 1 | 1 |
| ≥10 s | 49,751.3 | 1976 | 39.7 | 1.046 (0.988, 1.107) | 1.042 (0.975, 1.114) |
| Ventilator care | 141,921.8 | 694 | 4.9 | | |
| <10 s | 85,750.1 | 404 | 4.7 | 1 | 1 |
| ≥10 s | 56,171.7 | 290 | 5.2 | 1.095 (0.942, 1.274) | 1.136 (0.947, 1.363) |

**Note:** All subjects were 66 years old at screening. The incidence was presented as patients per 1000 persons/year. Data were analyzed using Cox regression models and are presented as adjusted hazard ratios (95% confidence intervals).

[a] Multivariable model included sex, smoking status, body mass index, unipedal balance test, regular physical activity, baseline cognitive function, depressive symptoms, and Charlson comorbidity index.

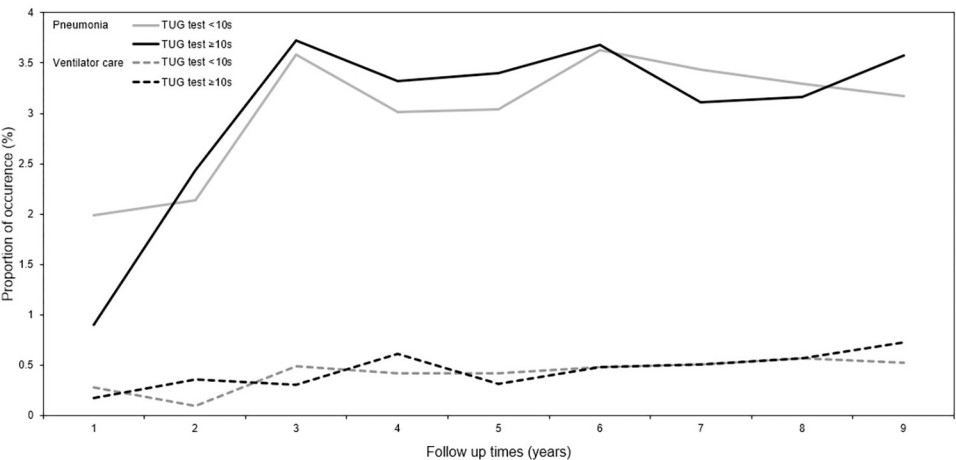

**Fig 2. Event occurrence of pneumonia and ventilator care during the follow-up period.** Abbreviations: TUG, timed up-and-go.

adults, the TUG test showed a statistical significant association with a history of falls [35]. Furthermore, a study examining the association between TUG testing and total mortality in people aged 65 and older found that there were significant associations between higher TUG scores (slower times) and increased all-cause mortality [36]. In addition, the TUG has demonstrated utility in predicting nursing home admissions [37], subsequent dementia [27] and even future cardiovascular events [25, 27, 37]. Despite the aforementioned studies on the usefulness of the TUG test, there are no studies on whether the TUG test can be used as a predictive tool for pneumonia occurrence.

In our study, the TUG test showed limited predictive value for subsequent pneumonia and ventilator care in our study. There are several possible explanations for the poor correlation. First, the population differed from other studies, including less healthy, lower-functioning older people. Our study included community-dwelling older adults and excluded those with stroke, Parkinson's disease, and dementia. Although the TUG test is a useful screening tool for sarcopenia, it may have limited ability to predict disease occurrence in community-dwelling elderly persons. Previous studies have demonstrated that TUG is not useful for predicting falls in healthy older adults and is not recommended as a tool to detect fall risk in healthy, high-functioning older people [38, 39]. Although these studies suggest that the predictive power of the TUG test may not be high in high-functioning older adults, there are no studies of pneumonia and the predictive power of the TUG test, and further research is needed beyond our study. The second issue is the cuff value. An accurate cutoff value for predicting sarcopenia using the TUG test has not been clearly established in community-dwelling elderly persons. A previous study showed that a cutoff value of 10.85 s or more was suggested for predicting sarcopenia with the TUG test in elderly hospitalized patients, and comparable cutoff values (10 s) were applied in our study [15]. However, in the aforementioned study conducted in inpatient settings [15], the cutoff value may differ from that of community-dwelling individuals [40]. In addition, the cutoff value (10–13 s) suggested in previous studies is not for screening for subsequent pneumonia but for fall risk in older adults [41, 42]. Therefore, further studies are needed to determine the cutoff value of TUG to predict subsequent pneumonia. Third, the population in previous research was commonly Caucasian [43], whereas our population was Asian. Since sarcopenia can be affected by various factors, such as ethnic or geographic factors [44–47], appropriate cutoff values may be necessary for Asian populations [48].

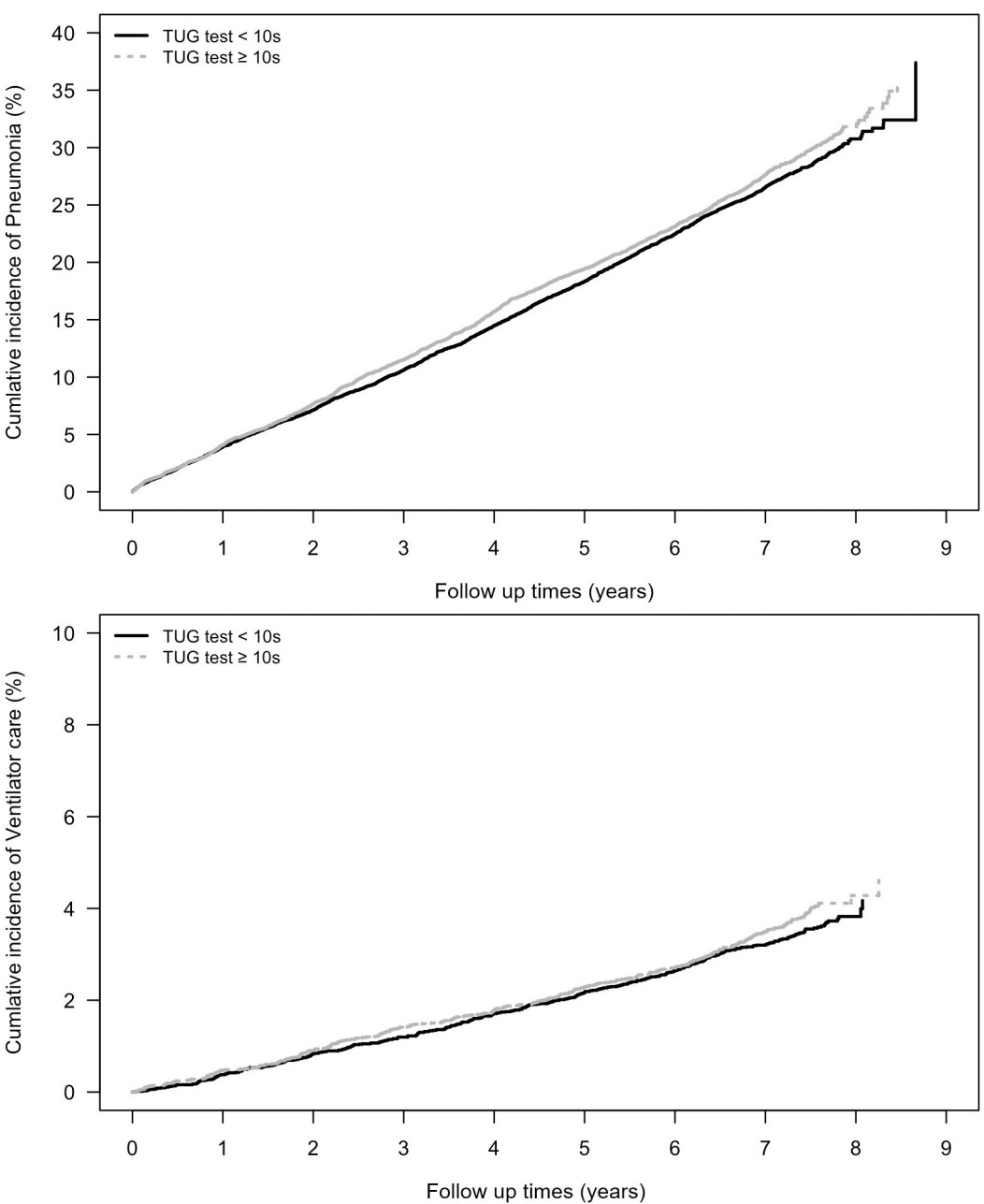

**Fig 3.** A) Cumulative incidence trend graph of pneumonia and B) ventilator care during the follow-up period.
**Abbreviations**: TUG, timed up-and-go.

In addition, this study evaluated the association between known risk factors and the occurrence of pneumonia. In our study, low body mass index [49, 50], smoking [51–53], and the presence of comorbid conditions [54–56] (presented as CCI score) were associated with subsequent pneumonia in old age, which has been confirmed in many previous studies. Regarding depressive symptoms [57], the presence of depression may confer an increased risk of infection and can be a risk factor for pneumonia [58]. In cognitive function, cognitive decline is thought to increase the risk of pneumonia due to reduced understanding of personal hygiene [59, 60],

so we included cognitive decline as a covariate in our analysis. However, unlike other studies, we did not find an association between cognitive decline and pneumonia [61].

Another interesting factor is the unipedal balance test, a functional test that assesses balance ability, such as the TUG test. In our study, an abnormal unipedal test was associated with subsequent pneumonia and ventilator care, whereas the TUG test was not associated with the outcomes. These findings suggest that the unipedal balance test may be useful. Previous research demonstrated that the TUG test revealed high specificity (74%) but low sensitivity (31%) for the prediction of falls [38] and other functional tests, such as the single-leg stance test, Berg Balance Scale, and the Five times sit to stand test showed promising results [62, 63]. Hurvitz et al. also proved that the sensitivity (91%) and specificity (71%) of the unipedal balance test were higher than those of the TUG test in predicting fall risk [64]. Therefore, it may be optimal to use the unipedal balance test rather than the TUG test to predict subsequent pneumonia, but further studies are needed to confirm these assumptions. Future studies could explore the synergistic effects of combining multiple balance tests to achieve more incremental and detailed results.

A limitation of this study is that pneumonia occurrence and ventilator care events are solely defined by the information in the KNHIS claim data [65]. Since the code was created to claim health insurance services to participants, the possibility that this code did not reflect the exact health status of the patients could not be excluded. Additionally, limited diagnostic items in the National Health Insurance Service-Senior Cohort (NHIS-Senior) database may be a limitation of the study. Sarcopenia diagnosis is based on a comprehensive evaluation of abnormalities in muscle mass, strength, and physical function, but the NHIS-Senior database only includes the TUG test and the unipedal test as balance assessment items for seniors, it was not possible to evaluate other diagnostic tests for muscle mass or strength. Consequently, there is a possibility that the slow TUG test results did not accurately reflect degree of sarcopenia. Another limitation is the appropriate cutoff value problem. As stated earlier, it is questionable whether applying the cutoff value (10 s) of the TUG test to the community-dwelling Korean elderly was appropriate. Next, the various conditions affecting TUG results were not explicitly evaluated. The TUG test protocol in the NSPTA manual does not designate the exact shape of the chair or walking pace, which can affect the TUG results [66]. However, many studies do not regulate the chair shape [42, 67], and there is some opinion that the shape of the chair does not affect the TUG test results [68].

However, this study included a large, nationally representative sample of the elderly population. This is the first study to identify the association between the TUG test and the risk of subsequent pneumonia and ventilator care. Our study evaluated the clinical characteristics of community-dwelling elderly people in real life, not in research circumstances. Despite these limitations, the aforementioned strengths make this study more meaningful.

## Conclusion

Our study suggests that there was no association with the risk of subsequent pneumonia occurrence or ventilator care in the slow TUG group ($\geq$10 s) when compared with the normal TUG group (<10 s). Our findings suggest that the TUG test may not be useful for predicting pneumonia occurrence in community-dwelling elderly individuals. Further studies are needed to identify additional functional tools for sarcopenia associated with future pneumonia occurrences.

## Author Contributions

**Conceptualization:** Hyo Jin Lee, Hyun Woo Lee, Jung-Kyu Lee, Eun Young Heo, Deog Kyeom Kim, Tae Yun Park.

**Data curation:** Hyo Jin Lee, Hyun Woo Lee, Eun Young Heo, Deog Kyeom Kim, Tae Yun Park.

**Formal analysis:** Hyo Jin Lee, Jung-Kyu Lee, Tae Yun Park.

**Investigation:** Hyo Jin Lee, Eun Young Heo, Tae Yun Park.

**Methodology:** Hyo Jin Lee, Sohee Oh, Jung-Kyu Lee, Eun Young Heo, Deog Kyeom Kim, Tae Yun Park.

**Project administration:** Hyo Jin Lee, Tae Yun Park.

**Resources:** Sohee Oh, Tae Yun Park.

**Software:** Sohee Oh, Tae Yun Park.

**Supervision:** Hyo Jin Lee, Hyun Woo Lee, Jung-Kyu Lee, Eun Young Heo, Deog Kyeom Kim, Tae Yun Park.

**Visualization:** Hyo Jin Lee, Tae Yun Park.

**Writing – original draft:** Hyo Jin Lee, Tae Yun Park.

**Writing – review & editing:** Hyo Jin Lee, Tae Yun Park.

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
