## [Decision Letter · Decision Letter 0]

14 Jun 2023

PONE-D-22-30626Association Between Timed Up-and-Go Test and Subsequent Pneumonia: A Cohort studyPLOS ONE

Dear Dr. Park,

Thank you for submitting your manuscript to PLOS ONE. After careful consideration, we feel that it has merit but does not fully meet PLOS ONE’s publication criteria as it currently stands. Therefore, we invite you to submit a revised version of the manuscript that addresses the points raised during the review process.

ACADEMIC EDITOR:  This study included a longitudinal cohort with sufficient follow up time. However, major concerns have been raised by the reviewers that must to be addressed and justified in order to have a clear picture to make a decision. Choosing the cutoff score for TUG was not justified. You can either justify or calculate the appropriate cutoff score for TUG using the ROC and the AUC because populations differ in some variables such as physical performance. 

We look forward to receiving your revised manuscript.

Kind regards,

Aqeel M Alenazi

Academic Editor

PLOS ONE

Journal Requirements:

Please include your amended statements within your cover letter; we will change the online submission form on your behalf."

Additional Editor Comments:

This study included a longitudinal cohort with sufficient follow up. However, some concerns have been raised by reviewers that need to be addressed and justified in order to have a clear picture to make a decision. Choosing the cutoff score for TUG was not justified. You can either justify or calculate the appropriate cutoff score for TUG using the ROC and the AUC because populations differ in some variables such as physical performance.

Reviewers' comments:

Reviewer's Responses to Questions

**Comments to the Author**

1. Is the manuscript technically sound, and do the data support the conclusions?

Reviewer #1: Yes

Reviewer #2: Partly

Reviewer #3: Yes

2. Has the statistical analysis been performed appropriately and rigorously? 

Reviewer #1: Yes

Reviewer #2: I Don't Know

Reviewer #3: N/A

3. Have the authors made all data underlying the findings in their manuscript fully available?

Reviewer #1: Yes

Reviewer #2: Yes

Reviewer #3: No

4. Is the manuscript presented in an intelligible fashion and written in standard English?

Reviewer #1: Yes

Reviewer #2: Yes

Reviewer #3: Yes

5. Review Comments to the Author

Reviewer #1: the TUG implemented to assess balance and functional performance following pneumonia could have been added up or compared with other balance tests like Berg balance scale or Five times sit to stand test to get better progressive results.

Reviewer #2: Thank you for considering me to review this manuscript.

Overall, the research question addressed in the study doesn't look much relevant.

Objectives of the study are inconsistently written at various places.

How can TUG be considered a clear indicator of sarcopenia?Though there exist a link between the two but solely TUG cannot be considered to be measure of sarcopenia. It is a test to measure mobility in elderly. Sarcopenia can only be ascertained by detailed muscle structure and function assessment.

Covariates considered in the present study aren't appropriate. Why do you cognition levels can cause pneumonia? It is taken as a potential covariate in the present study.

How the Sample size calculation for the present study was done?? There is no mention about it in the manuscript.

Conducting a multivariate logistic regression analysis in which predictive ability of TUG for pneumonia can be considered for in depth analysis on the topic.

Reviewer #3: Dear authors,

The manuscript you have written is of very good quality. Your cohort study presents interesting information. I do have a few comments and questions that I would like you to address.

Study population:

Line 109: please clarify what do you mean by: “This figure represents 10% of….”

Independent variable:

You need to include a reference that provides information about the TUG (Timed Up and Go) test.

You need to talk here about the justification to choose 10s as a cutoff point for TUG test.

According to the International Classification of Functioning, Disability, and Health (ICF), impairment is defined as "problems in body function and structure." Consequently, I recommend replacing the term "impaired TUG" with "slow TUG" when describing TUG results that exceed the cutoff point. Additionally, it is not typical to assume that an individual has “gait impairment” or “abnormality" solely based on their TUG score surpassing a specific cutoff point. So, I suggest to use another description such as slow performance.

Outcome variable:

It is good if you cite a reference for the coding system.

Covariates:

Line 159: specify which questionnaire?

Line 160-163: Are there any references for selecting those cutoff points?

It is generally advisable to include references for the outcome measures utilized in this study to substantiate their reliability and validity.

Line 181: this is the first place to mention the CCI. You need to introduce it earlier under covariates section.

Results:

Table 1:

Can you confirm if there is a significant difference between groups for age and BMI.

Regarding smoking status, be consistent with naming categories (e.g., non-smoker and ever-smoker, past and former)

In table 1 first column: what do you mean by “ADL-impaired” outcome and how was it measured?

Table 2:

It is not mentioned/clear in the table whether you categorized subjects based on their BMI.

Line 259- correct: “…a higher risk of pneumonia occurrence …” to “…a higher risk of ventilator events…”

Is there any justification to have the end of follow up in 2015?

Line 251: is it 39.5/1000 or 39.7/1000

Table 4: can you explain what the second column refers to?

Line 283: ex-smoker or non-smoker?

Line 288: “..has been widely used in many studies…” cite examples on these studies

Line 292:

In this paragraph, it is obvious that you are discussing the possible explanations for the limited predictive ability of TUG, and that your sample may be healthier than the samples of other studies or differ due to ethnic and geographic factors. These factors may affect your results, but you are comparing them to studies that investigated different outcomes.

Line 327- it may be better to say: to confirm these assumptions instead of results.

Last point, you may need to double check on the journal reference style.

6. PLOS authors have the option to publish the peer review history of their article (what does this mean?). If published, this will include your full peer review and any attached files.

Reviewer #1: **Yes: **PREM KUMAR BHOJARA

Reviewer #2: No

Reviewer #3: No

---

## [Author Response · Author response to Decision Letter 0]

14 Aug 2023

Reviewers' comments:

Reviewer's Responses to Questions

Comments to the Author

1. Is the manuscript technically sound, and do the data support the conclusions?

Reviewer #1: Yes

Reviewer #2: Partly

Reviewer #3: Yes

2. Has the statistical analysis been performed appropriately and rigorously?

Reviewer #1: Yes

Reviewer #2: I Don't Know

Reviewer #3: N/A

3. Have the authors made all data underlying the findings in their manuscript fully available?

Reviewer #1: Yes

Reviewer #2: Yes

Reviewer #3: No

4. Is the manuscript presented in an intelligible fashion and written in standard English?

Reviewer #1: Yes

Reviewer #2: Yes

Reviewer #3: Yes

5. Review Comments to the Author

Reviewer #1: 

◆ The TUG implemented to assess balance and functional performance following pneumonia could have been added up or compared with other balance tests like Berg balance scale or Five times sit to stand test to get better progressive results.

Response: Thanks for your valuable comment. We appreciate the reviewer's suggestion to consider including additional balance tests such as the Berg Balance Scale or Five Times Sit to Stand Test in our study. These tests indeed provide valuable information regarding balance and functional performance in the elderly population.

We reviewed all the items of database again to add the tests you mentioned and analyze them, but the National Health Insurance Service-Senior Cohort (NHIS-Senior) database did not provide data on other balance tests such as the Berg balance scale or five times sit to stand test other than the TUG test. Therefore, it was not possible to evaluate other balance tests such as Berg Balance Scale or Five Times Sit to Stand Test. 

It is unfortunate that the tests mentioned are not available. Instead, we have described the usefulness of mentioned test as reference and limitations of NHIS-senior database in the discussion. 

We revised Discussion as follows; 

Discussion: Line 371, 375-377, 381-387

Reviewer #2: Thank you for considering me to review this manuscript.

Overall, the research question addressed in the study doesn't look much relevant.

Objectives of the study are inconsistently written at various places.

◆ How can TUG be considered a clear indicator of sarcopenia? Though there exist a link between the two but solely TUG cannot be considered to be measure of sarcopenia. It is a test to measure mobility in elderly. Sarcopenia can only be ascertained by detailed muscle structure and function assessment.

Response: Thanks for your valuable comment. I completely agree that the TUG test is not the one definitive tool for diagnosing sarcopenia. A combination of other tests usually used to diagnose sarcopenia.

However, realistically, it is almost impossible to obtain a big data set (more than 10,000 people) that all the items with calf circumference, handgrip strength, physical performance test, DEXA or BIA test data, so as an alternative, we analyzed the National Health Insurance Service-Senior Cohort, which has data from the TUG test performed by seniors aged 66 and older in South Korea.

Since the TUG test involves getting up from a chair, walking 3 meters, and returning, it is thought to be useful as a screening test for patients with sarcopenia, and is reflective of functional sarcopenia, so we chose this test for our study.

According to Korean Working Group on Sarcopenia Guideline, TUG test may simplify the diagnostic steps for sarcopenia in clinical settings with a high prevalence of sarcopenia [Reference- Ann Geriatr Med Res 2023;27(1):9-21]. There are studies to support that the TUG test can be feasible assessment tool for sarcopenia. The TUG test effectively screened for sarcopenia among 332 elderly individuals with high sensitivity (88.9%) and negative predictive value (93.2%) in a southern Brazilian city, particularly those with good physical and cognitive abilities. Another study also demonstrated that TUG test predicted sarcopenia with a sensitivity of 67% and a specificity of 88.7% in elderly patients. 

Considering that sarcopenia evaluation items include muscle mass [Reference-Age Ageing. 2010;39(4):412–23], strength, and physical performance, and one of the physical performance evaluation items is the TUG test, the TUG test is a simple test that can be used even in a limited resource environment [Reference-Clinics (Sao Paulo). 2015;70(5):369-72]. A cross-sectional study showed that muscle mass and physical performance were associated with gait-speed and TUG tests [BMC Geriatrics. 2014;14(13):1–7.]. In this study, elderly women with reduced muscle mass had poor physical performance with slow TUG tests over 10.85 seconds.

The comments you mentioned are reasonable, and we have supplemented the above reference and limitation in the Introduction and Discussion parts. Thank you very much. 

We revised Introduction and Discussion as follows; 

Introduction: line 84-95

Discussion: line 381-387 

◆ Covariates considered in the present study aren't appropriate. Why do you cognition levels can cause pneumonia? It is taken as a potential covariate in the present study.

Response: We appreciate the reviewer's valuable comment. We used cognitive function as a covariate for two reasons. 

First, there is a direct relationship between the severity of cognitive impairment and increased gait abnormalities. Gait impairment is known to occur in mild cognitive impairment as well as Alzheimer's disease, and it was thought that this could potentially affect the performance on the TUG test [Reference- Neurosci Biobehav Rev 2007; 31: 485-97]. Previous studies also demonstrated that TUG test is associated with cognitive function, quantitatively [. J Am Geriatr Soc. 2011;59(11):2188–9., Gerontology. 2011;57(3):203–10., Phys Ther. 2011;91(8):1198–207., J Am Geriatr Soc. 2012;60(9):1681–6.]. Therefore, if the slowed gait was due to cognitive function and not to sarcopenia, we would need to correct for that, so we included cognitive function as a covariate in the analysis.

Second, there is a relationship between cognitive function and pneumonia. Previous studies reported that cognitive decline in elderly patients increased their risk of aspiration pneumonia, likely due to factors like a deteriorating oral environment caused by reduced personal care, difficulty in understanding how to use objects, and reduced hand dexterity [Oral Investig. 2018, 22, 2575–2580., J. Neurosci. Res. 2021, 99, 518–528.]. Also, A longitudinal analysis of cognitive function and pneumonia incidence over a 10-year period found that the even small subclinical cognitive changes have been shown to increase the risk of hospitalization for pneumonia [Reference- Am J Respir Crit Care Med. 2013 Sep 1;188(5):586-92.]. 

In conclusion, the inclusion of cognitive levels as a covariate in our analysis was based on the above rationale. However, as mentioned your comment, we felt that the description of why we chose cognitive level as a covariate was lacking, so we have added a reference to the rationale and a sentence to explain. 

We revised Discussion as follows; 

Discussion: line 361-364

◆How the Sample size calculation for the present study was done?? There is no mention about it in the manuscript.

Response: NHIS-Senior database consisted of approximately 5.5 million seniors aged 60 and older who were still eligible for health insurance and medical benefits at the end of December 2002. The sampling method was a simple random sample of approximately 550,000 people, which is 10% of this population. 

We revised Materials and methods as follows;

Materials and methods: lines 125-129

◆Conducting a multivariate logistic regression analysis in which predictive ability of TUG for pneumonia can be considered for in depth analysis on the topic.

Response: Thanks for the good point. As you mentioned, multivariate logistic regression for the association between TUG test and the occurrence of pneumonia itself can also be helpful. However, in the case of this study, since it was a cohort data with patients' data collected from 2002 and followed up from 2015, it was necessary to include the "time" item in the statistical analysis, and we used a Cox proportional hazard model to examine the association between TUG test and the time of first pneumonia (future pneumonia) occurrence. 

Reviewer #3: Dear authors,

The manuscript you have written is of very good quality. Your cohort study presents interesting information. I do have a few comments and questions that I would like you to address.

Study population:

◆ Line 109: please clarify what do you mean by: “This figure represents 10% of….”

Response: NHIS-Senior database consisted of approximately 5.5 million seniors aged 60 and older who were still eligible for health insurance and medical benefits at the end of December 2002. The sampling method was a simple random sample of approximately 550,000 people, which is 10% of this population. 

We revised Materials and methods as follows;

Material and Method: lines 125-129

Independent variable:

◆ You need to include a reference that provides information about the TUG (Timed Up and Go) test.

Response: Thanks for your comment. We include a reference regarding the TUG test. We revised methods and materials as follows. 

Material and Method (Independent variable): lines 160 (Reference 20, line 487-488) 

◆ You need to talk here about the justification to choose 10s as a cutoff point for TUG test.

According to the International Classification of Functioning, Disability, and Health (ICF), impairment is defined as "problems in body function and structure." Consequently, I recommend replacing the term "impaired TUG" with "slow TUG" when describing TUG results that exceed the cutoff point. Additionally, it is not typical to assume that an individual has “gait impairment” or “abnormality" solely based on their TUG score surpassing a specific cutoff point. So, I suggest to use another description such as slow performance.

Response: Thanks for your comment. We replace the term “impaired TUG” with “slow TUG” according to your opinion. We revised through manuscript.

Abstract: line 36, 37, 39, 40

Materials and methods: line 160

Results: line 228, 230, 231, 233, 285, 286, 287, 290, 293 

Discussion: line 316

Conclusion: line 403

Outcome variable:

◆ It is good if you cite a reference for the coding system.

Response: Thank you for your comments. We used diagnostic codes compatible to the ICD-10 coding and Healthcare Common Procedure Coding System codes provided by the Health Insurance Review and Assessment Service. 

We revised materials and methods and cited this as references as follows; Materials and methods: line 167 (Reference 21,22, line 489-494), 170 (Reference 23,24, line 495-503)

Covariates:

◆ Line 159: specify which questionnaire?

Response: Thanks for your comment. This is a survey questionnaire for the current smoking status of participants: (1) Never smoker, (2) Former smoker who has quit, (3) Current smoker. The questionnaire also includes inquiries regarding smoking history, both past and present, with regards to: (1) Smoking duration (1. <5 years, 2. 5-9 years, 3. 10-19 years, 4. 20-29 years, 5. 30 years or more), and (2) Daily cigarette consumption (1. <0.5 pack, 2. 0.5-1 pack, 3. 1-2 packs, 4. 2 packs or more). 

We revised Materials and methods as follow; 

Material and Methods (Covariate): line 177-182

◆ Line 160-163: Are there any references for selecting those cutoff points?

It is generally advisable to include references for the outcome measures utilized in this study to substantiate their reliability and validity.

Response: Thanks for your comments. We add the references of cutoff point. 

Material and Methods (Covariate): line 187 (Reference 25, 26, line 504-509) 

◆ Line 181: this is the first place to mention the CCI. You need to introduce it earlier under covariates section.

Response: Thanks for your comment. We revised Methods and materials as follows.

Material and Methods (Covariate): line 185

Results: line 210

◆ Table 1. Can you confirm if there is a significant difference between groups for age and BMI.

Regarding smoking status, be consistent with naming categories (e.g., non-smoker and ever-smoker, past and former)

Response: Thanks for your comment. We tried rerunning the statistical analysis on the age and BMI items, but the results are the same, and the p value is the same as the original result. 

 Total TUG test P-value

 <10s ≥10s 

Age (y) 66.06 ± 0.23 66.05 ± 0.22 66.07 ± 0.25 <0.0001

BMI, Kg/m2 (continuous) 24.25 ± 3.02 24.19 ± 2.93 24.34 ± 3.15 <0.0001

BMI, Kg/m2 (categorical) 

<18.5 442 (2.23%) 261 (2.18%) 181 (2.30%) 0.0002

≤18.5–<25 11,862 (59.90%) 7,259 (60.76%) 4,603 (59.60%) 

≤25–<30 6,837 (34.53%) 4,076 (34.12%) 2,761 (35.15%) 

≥30 661 (3.34%) 351 (2.94%) 310 (3.95%) 

Also, we corrected the naming categories. We revised Result as follows; 

Results: line 254-255, 257; Table 1 and line 239, 240; Table 2 and line 263, 264

◆ In table 1 first column: what do you mean by “ADL-impaired” outcome and how was it measured?

Response: Thanks for your comment. We revised Methods and materials as follows.

Results: line 187-190

◆ Table 2:

It is not mentioned/clear in the table whether you categorized subjects based on their BMI.

Response: Thanks for your comment. We considered BMI as a continuous variable rather than a categorical variable. In other words, since the hazard ratio of BMI for pneumonia occurrence in Table 2 is 0.98 and the P-value is 0.0003, it can be interpreted that the lower the BMI, the higher the risk of pneumonia in the univariable analysis. We mentioned it below the table. 

Results: line 257, Table 1 and line 239, Table 2 and line 263

◆ Line 259- correct: “…a higher risk of pneumonia occurrence …” to “…a higher risk of ventilator events…”

Response: Thanks for your comment. We revised Results as follows.

Results: line 293-294

◆ Is there any justification to have the end of follow up in 2015?

Response: Thanks for your comment. The National Health Insurance Service's senior cohort DB, which was built with data of seniors aged 60 or older to support research such as risk factors and prognostic analysis of geriatric diseases, was only made public from 2002 to 2015. Therefore, the period available for request to the National Health Insurance Corporation coincides with the period of our study. It is considered that it will be possible later to use the DB from 2016 to the latest.

◆ Line 251: is it 39.5/1000 or 39.7/1000

Response: Thanks for your comment. We revised Result as follows.

Results: line 286

◆ Table 4: can you explain what the second column refers to?

Response: Thanks for your question. The incidence of pneumonia and ventilator care is presented as patients per 1000 persons/year. This means that the number of patients who developed pneumonia or required ventilator care during the follow-up period (which was 13 years in this study) was divided by the total number of persons in the study cohort, and then multiplied by 1000 to obtain the incidence rate per 1000 persons per year. This is a common way of reporting incidence rates in epidemiological studies. We have previously presented as a note under the Table 4 as follows;

Results: line 299

Note: All subjects were 66 years old at screening. The incidence was presented as patients per 1000 persons/year. Data were analyzed using Cox regression models and are presented as adjusted hazard ratios (95% confidence intervals).

Line 283: ex-smoker or non-smoker?

Response: Thanks for your question. Ever-smokers were defined as former or current smokers excluding never smokers. We revised Results as following.

Results: line 254-255, 257, Table 1 and line 240, Table 2 and line 264

◆ Line 288: “..has been widely used in many studies…” cite examples on these studies

Response: Thanks for the valuable comment. This sentence aims to highlight the effectiveness of the TUG test, a useful tool for assessing functional impairment in the elderly population, in predicting various healthcare outcomes, including a history of falls [Reference-BMC Geriatr. 2007;7:1.], all-cause mortality [Reference-BMC Health Serv Res. 2017;17(1):22], nursing home admissions [Reference-Aging (Milano). 1996;8(4):271-6], onset of dementia [Reference-J Gerontol A Biol Sci Med Sci. 2018;73(9):1238-43.], and even future cardiovascular events [Reference-BMC Geriatr. 2020;20(1):111.]. Therefore, we've made detailed examples of those studies and added the references.

We revised Discussion as follows. 

Discussion: line 322-332 (Reference: 11-13, line 458-466, Reference: 28-32, line 513-527) 

Line 292:

In this paragraph, it is obvious that you are discussing the possible explanations for the limited predictive ability of TUG, and that your sample may be healthier than the samples of other studies or differ due to ethnic and geographic factors. These factors may affect your results, but you are comparing them to studies that investigated different outcomes.

Response: Thanks for the very helpful comments. I completely agree with the comments. As you mentioned, for an accurate comparison of outcomes, it would be valid to compare studies that associate TUG results with pneumonia incidence. However, there are no existing studies on the prediction of pneumonia and TUG test and there are no studies perfectly match our study. So, we toned down the discussion and added a sentence.

Discussion: line 341-343, line 381-387 

Line 327- it may be better to say: to confirm these assumptions instead of results.

Response: Thanks for your comment. We revised Discussion as follows.

Discussion: line 375-377

Last point, you may need to double check on the journal reference style.

Response: Thanks for your comment. We double checked on the journal reference style as Vancouver.

6. PLOS authors have the option to publish the peer review history of their article (what does this mean?). If published, this will include your full peer review and any attached files.

Do you want your identity to be public for this peer review? For information about this choice, including consent withdrawal, please see our Privacy Policy.

Reviewer #1: Yes: PREM KUMAR BHOJARA

Reviewer #2: No

Reviewer #3: No

---

## [Decision Letter · Decision Letter 1]

14 Sep 2023

PONE-D-22-30626R1Association Between Timed Up-and-Go Test and Subsequent Pneumonia: A Cohort studyPLOS ONE

Dear Dr. Park,

Thank you for submitting your manuscript to PLOS ONE. After careful consideration, we feel that it has merit but does not fully meet PLOS ONE’s publication criteria as it currently stands. Therefore, we invite you to submit a revised version of the manuscript that addresses the points raised during the review process.

ACADEMIC EDITOR: The remaining comment from the reviewer and editor has not been addressed clearly. Choosing the cutoff score for TUG of 10 s was not justified. You can either justify or calculate the appropriate cutoff score for TUG using the ROC and the AUC because populations differ in some variables such as physical performance.

We look forward to receiving your revised manuscript.

Kind regards,

Aqeel M Alenazi

Academic Editor

PLOS ONE

Journal Requirements:

Additional Editor Comments (if provided):

The remaining comment from the reviewer and editor has not been addressed clearly. Choosing the cutoff score for TUG of 10 s was not justified. You can either justify or calculate the appropriate cutoff score for TUG using the ROC and the AUC because populations differ in some variables such as physical performance.

Reviewers' comments:

Reviewer's Responses to Questions

**Comments to the Author**

1. If the authors have adequately addressed your comments raised in a previous round of review and you feel that this manuscript is now acceptable for publication, you may indicate that here to bypass the “Comments to the Author” section, enter your conflict of interest statement in the “Confidential to Editor” section, and submit your "Accept" recommendation.

Reviewer #1: All comments have been addressed

Reviewer #3: All comments have been addressed

2. Is the manuscript technically sound, and do the data support the conclusions?

Reviewer #1: Yes

Reviewer #3: Yes

3. Has the statistical analysis been performed appropriately and rigorously? 

Reviewer #1: Yes

Reviewer #3: (No Response)

4. Have the authors made all data underlying the findings in their manuscript fully available?

Reviewer #1: Yes

Reviewer #3: No

5. Is the manuscript presented in an intelligible fashion and written in standard English?

Reviewer #1: Yes

Reviewer #3: Yes

6. Review Comments to the Author

Reviewer #1: Manuscript is written is of good relevant information. The importance of understanding the relation between sarcopenia, TUG, pneumonia, balance test is expalined.

Reviewer #3: Dear authors,

I appreciate your efforts in taking into account the feedback provided by the reviewers, and for diligently addressing the suggestions and comments. This commitment enhanced the quality of the manuscript.

But one final comment remains, is to provide a clear justification or reference for choosing the cutoff point for the TUG test.

7. PLOS authors have the option to publish the peer review history of their article (what does this mean?). If published, this will include your full peer review and any attached files.

Reviewer #1: **Yes: **PREM KUMAR BHOJARA

Reviewer #3: **Yes: **Sumayeh Abujaber

---

## [Author Response · Author response to Decision Letter 1]

26 Oct 2023

ACADEMIC EDITOR: The remaining comment from the reviewer and editor has not been addressed clearly. Choosing the cutoff score for TUG of 10 s was not justified. You can either justify or calculate the appropriate cutoff score for TUG using the ROC and the AUC because populations differ in some variables such as physical performance.

Response) Thank you for your valuable comments. Our study used data from a special health-screening program called the National Screening Program for Transitional Ages (NSPTA) provided by the National Health Insurance Service-Senior Cohort (NHIS-Senior) database of the Korean National Health Insurance (KNHI) service. The NSPTA includes a questionnaire regarding mental/cognitive function and depression screening, as well as physical function tests (TUG test, unipedal balance test) that cover common problems of the elderly, such as frailty. 

In the Korean NSPTA, the TUG test cutoff is defined as normal at 10 seconds or less and abnormal at 10 seconds or more based on expert opinion. In addition, the cutoff of 9-10 seconds is used as a cutoff in various populations, and the cutoff of 10 seconds is commonly used in ethnically similar Asian countries such as Japan and Singapore [Reference- J Am Med Dir Assoc. 2021 Aug;22(8):1640-1645., Phys Ther 2017;97(4):417-24.] In fact, most Korean studies that have analyzed the association between TUG test and clinical outcomes using data from the National Screening Program for Transitional Ages in Korea have also used a cutoff value of 10 seconds for the TUG test [Reference- Bone. 2019 Oct;127:474-481., BMC Geriatr . 2020 Mar 19;20(1):111., J Korean Med Sci. 2020 Jan 20; 35(3): e25., J Gerontol A Biol Sci Med Sci. 2018;73(9):1238-43.].

Furthermore, our study focused on investigating the correlation between a slow TUG result and the occurrence of future pnuemonia, rather than setting a cut off value for the TUG test in detecting pneumonia. Since the cut off suggested by Korean NAPTA and most of the studies using the Korean NAPTA data mentioned above are based on a 10-second cut off, we believe that it is appropriate to apply a 10-second cut off.

Therefore, as you commented, we have further described and written the rationale for setting the cutoff in the Independent variable section as follows. (Line 157-164). We would also like to express our sincere thanks to the editor and reviewers for their helpful comments to improve the paper. 

Materials and methods

…

Independent variable 

…

The cut off value of the TUG test varies depending on the study, but the cut off value of 9-10 seconds is used in various population groups (20, 21) , and the cut off value of 10 seconds is also used in Asian populations such as Japan and Singapore (22, 23). In fact, most studies investigating the association of the TUG test with clinical outcomes such as dementia, fractures, and functional disability in Korean populations have used a 10-second cut off value (24-27). Based on these existing studies and expert opinions, the Korean NSPTA defines a TUG test of less than 10 seconds as normal, so in our study, participants were divided into two groups based on their TUG test (<10 seconds as normal and ≥10 seconds as slow gait).

Reviewers' comments:

Reviewer's Responses to Questions

Comments to the Author

1. If the authors have adequately addressed your comments raised in a previous round of review and you feel that this manuscript is now acceptable for publication, you may indicate that here to bypass the “Comments to the Author” section, enter your conflict of interest statement in the “Confidential to Editor” section, and submit your "Accept" recommendation.

Reviewer #1: All comments have been addressed

Reviewer #3: All comments have been addressed

2. Is the manuscript technically sound, and do the data support the conclusions?

Reviewer #1: Yes

Reviewer #3: Yes

3. Has the statistical analysis been performed appropriately and rigorously?

Reviewer #1: Yes

Reviewer #3: (No Response)

4. Have the authors made all data underlying the findings in their manuscript fully available?

Reviewer #1: Yes

Reviewer #3: No

5. Is the manuscript presented in an intelligible fashion and written in standard English?

Reviewer #1: Yes

Reviewer #3: Yes

6. Review Comments to the Author

Reviewer #1: Manuscript is written is of good relevant information. The importance of understanding the relation between sarcopenia, TUG, pneumonia, balance test is expalined.

Reviewer #3: Dear authors,

I appreciate your efforts in taking into account the feedback provided by the reviewers, and for diligently addressing the suggestions and comments. This commitment enhanced the quality of the manuscript.

But one final comment remains, is to provide a clear justification or reference for choosing the cutoff point for the TUG test.

Response) Thank you for your valuable comments. Our study used data from a special health-screening program called the National Screening Program for Transitional Ages (NSPTA) provided by the National Health Insurance Service-Senior Cohort (NHIS-Senior) database of the Korean National Health Insurance (KNHI) service. The NSPTA includes a questionnaire regarding mental/cognitive function and depression screening, as well as physical function tests (TUG test, unipedal balance test) that cover common problems of the elderly, such as frailty. 

In the Korean NSPTA, the TUG test cutoff is defined as normal at 10 seconds or less and abnormal at 10 seconds or more based on expert opinion. In addition, the cutoff of 9-10 seconds is used as a cutoff in various populations, and the cutoff of 10 seconds is commonly used in ethnically similar Asian countries such as Japan and Singapore [Reference- J Am Med Dir Assoc. 2021 Aug;22(8):1640-1645., Phys Ther 2017;97(4):417-24.] In fact, most Korean studies that have analyzed the association between TUG test and clinical outcomes using data from the National Screening Program for Transitional Ages in Korea have also used a cutoff value of 10 seconds for the TUG test [Reference- Bone. 2019 Oct;127:474-481., BMC Geriatr . 2020 Mar 19;20(1):111., J Korean Med Sci. 2020 Jan 20; 35(3): e25., J Gerontol A Biol Sci Med Sci. 2018;73(9):1238-43.].

Therefore, as you commented, we have further described and written the rationale for setting the cutoff in the Independent variable section as follows. (Line 157-164)

We would also like to express our sincere thanks to the editor and reviewers for their helpful comments to improve the paper. 

Materials and methods

…

Independent variable 

…

The cut off value of the TUG test varies depending on the study, but the cut off value of 9-10 seconds is used in various population groups (20, 21) , and the cut off value of 10 seconds is also used in Asian populations such as Japan and Singapore (22, 23). In fact, most studies investigating the association of the TUG test with clinical outcomes such as dementia, fractures, and functional disability in Korean populations have used a 10-second cut off value (24-27). Based on these existing studies and expert opinions, the Korean NSPTA defines a TUG test of less than 10 seconds as normal, so in our study, participants were divided into two groups based on their TUG test (<10 seconds as normal and ≥10 seconds as slow gait).

7. PLOS authors have the option to publish the peer review history of their article (what does this mean?). If published, this will include your full peer review and any attached files.

Do you want your identity to be public for this peer review? For information about this choice, including consent withdrawal, please see our Privacy Policy.

Reviewer #1: Yes: PREM KUMAR BHOJARA

Reviewer #3: Yes: Sumayeh Abujaber

---

## [Decision Letter · Decision Letter 2]

12 Dec 2023

Association Between Timed Up-and-Go Test and Subsequent Pneumonia: A Cohort study

PONE-D-22-30626R2

Dear Dr. Park,

We’re pleased to inform you that your manuscript has been judged scientifically suitable for publication and will be formally accepted for publication once it meets all outstanding technical requirements.

Kind regards,

Aqeel M Alenazi

Academic Editor

PLOS ONE

Additional Editor Comments (optional):

Authors addressed all comments and suggestions

Reviewers' comments:

Reviewer's Responses to Questions

**Comments to the Author**

1. If the authors have adequately addressed your comments raised in a previous round of review and you feel that this manuscript is now acceptable for publication, you may indicate that here to bypass the “Comments to the Author” section, enter your conflict of interest statement in the “Confidential to Editor” section, and submit your "Accept" recommendation.

Reviewer #3: All comments have been addressed

2. Is the manuscript technically sound, and do the data support the conclusions?

Reviewer #3: Yes

3. Has the statistical analysis been performed appropriately and rigorously? 

Reviewer #3: N/A

4. Have the authors made all data underlying the findings in their manuscript fully available?

Reviewer #3: No

5. Is the manuscript presented in an intelligible fashion and written in standard English?

Reviewer #3: Yes

6. Review Comments to the Author

Reviewer #3: Authors addressed all comments. Thank you for addressing the comment and providing clear rationale for the used cutoff point.

7. PLOS authors have the option to publish the peer review history of their article (what does this mean?). If published, this will include your full peer review and any attached files.

Reviewer #3: **Yes: **Sumayeh Abujaber

---

## [Editor Report · Acceptance letter]

27 Dec 2023

PONE-D-22-30626R2 

PLOS ONE

Dear Dr. Park, 

I'm pleased to inform you that your manuscript has been deemed suitable for publication in PLOS ONE. Congratulations! Your manuscript is now being handed over to our production team.

Kind regards, 

on behalf of

Dr. Aqeel M Alenazi 

Academic Editor

PLOS ONE